# UniDrive: Towards Universal Driving Perception Across Camera Configurations

**Ye Li**[1][*]   **Wenzhao Zheng**[2][†]   **Xiaonan Huang**[1]   **Kurt Keutzer**[2]
[1]University of Michigan, Ann Arbor    [2]University of California, Berkeley
https://wzzheng.net/UniDrive

## Abstract

Vision-centric autonomous driving has demonstrated excellent performance with economical sensors. As the fundamental step, 3D perception aims to infer 3D information from 2D images based on 3D-2D projection. This makes driving perception models susceptible to sensor configuration (e.g., camera intrinsics and extrinsics) variations. However, generalizing across camera configurations is important for deploying autonomous driving models on different car models. In this paper, we present **UniDrive**, a novel framework for vision-centric autonomous driving to achieve universal perception across camera configurations. We deploy a set of unified virtual cameras and propose a ground-aware projection method to effectively transform the original images into these unified virtual views. We further propose a virtual configuration optimization method by minimizing the expected projection error between original and virtual cameras. The proposed virtual camera projection can be applied to existing 3D perception methods as a plug-and-play module to mitigate the challenges posed by camera parameter variability, resulting in more adaptable and reliable driving perception models. To evaluate the effectiveness of our framework, we collect a dataset on CARLA by driving the same routes while only modifying the camera configurations. Experimental results demonstrate that our method trained on one specific camera configuration can generalize to varying configurations with minor performance degradation.

## 1 Introduction

Vision-centric autonomous driving has gained significant traction (Wang et al., 2023a) due to its ability to deliver high-performance perception using economical sensors like cameras. At the core of this approach lies 3D perception (Liu et al., 2022), which reconstructs 3D spatial information from 2D images via 2D-3D lift transform (Philion & Fidler, 2020). This transform is critical for enabling vehicles to understand their environment, detect objects, and navigate safely. Previous works (Huang et al., 2021; Xie et al., 2022; Reading et al., 2021; Li et al., 2022; Zhou & Krähenbühl, 2022; Zeng et al., 2024; Lu et al., 2022; Huang & Huang, 2022; Liu et al., 2023a) have achieved remarkable 3D perception ability by utilizing Bird's Eye View (BEV) representations to process 2D-3D lift. Recently, many vision-based 3D occupancy prediction methods (Huang et al., 2023; Wei et al., 2023; Huang et al., 2024a;b; Zhao et al., 2024) further improved the understanding of dynamic and cluttered driving scenes, pushing the boundaries of the research domain. As a result, vision-based systems have become one of the primary solutions for scalable autonomous driving.

Despite the exciting development of the state-of-the-art vision-based autonomous driving (Liu et al., 2024; Zong et al., 2023; Zheng et al., 2024b;a; Hu et al., 2023; Jiang et al., 2023a), a critical limitation still remains: the sensitivity of these models to variations in camera configurations, including intrinsics and extrinsics. Autonomous driving models typically rely on well-calibrated sensor setups, and even slight deviations in camera parameters across different vehicles or platforms can significantly degrade performance (Wang et al., 2023b). As illustrated in Figure 1a, this lack of robustness to sensor variability makes it challenging to transfer perception models between different vehicle platforms without extensive retraining or manual adjustment. This variation necessitates

---

[*]Work done while visiting UC Berkeley.
[†]Corresponding author.

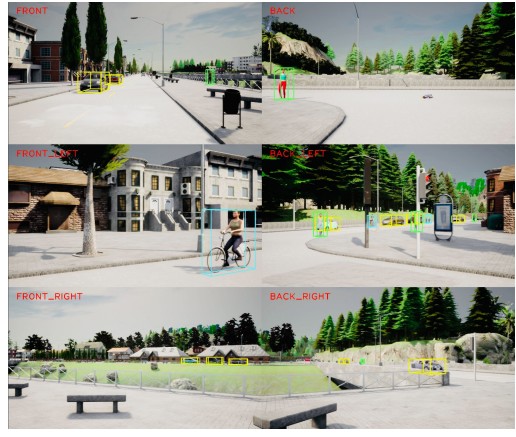 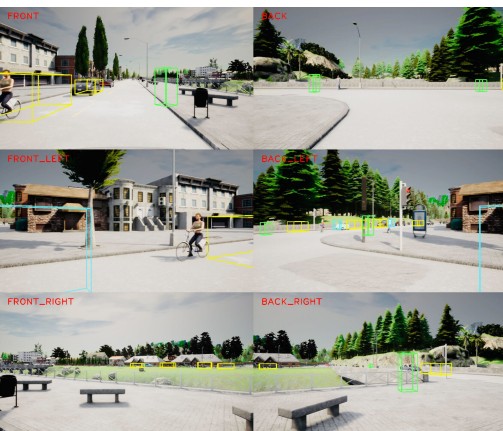

(a) Deploy on *same* camera configurations: Succeed!  (b) Deploy on *different* camera configurations: Fail!

Figure 1: Comparison of deploying perception models on the same and distinct configurations.

training separate models for each vehicle, which consumes a significant amount of computational resources. Thus, achieving generalization across camera configurations is essential for the practical deployment of vision-centric autonomous driving.

In this paper, we address two key questions surrounding generalizable driving perception: *1) How can we construct a unified framework that enables perception models to generalize across different multi-camera parameters? 2) How can we further optimize the generalization of perception models to ensure robust performance across varying multi-camera configurations?*

To achieve this, we introduce **UniDrive**, a novel framework designed to address the challenge of generalizing perception models across multi-camera configurations. This framework deploys a set of unified virtual camera spaces and leverages a ground-aware projection method to transform original camera images into these unified virtual views. Additionally, we propose a virtual configuration optimization strategy that minimizes the expected projection error between the original and virtual cameras, enabling consistent 3D perception across diverse setups. Our framework serves as a plug-and-play module for existing 3D perception methods, improving their robustness to camera parameter variability. We validate our framework in CARLA by training and testing models on different camera configurations, demonstrating that our approach significantly reduces performance degradation while maintaining adaptability across diverse sensor setups. To summarize, we make the following key contributions in this paper:

- To the best of our knowledge, **UniDrive** presents the first comprehensive framework designed to generalize vision-centric 3D perception models across diverse camera configurations.

- We introduce a novel strategy that transforms images into a unified virtual camera space, enhancing robustness to camera parameter variations.

- We propose a virtual configuration optimization strategy that minimizes projection error, improving model generalization with minimal performance degradation.

- We contribute a systematic data generation platform along with a 160,000 frames multi-camera dataset, and benchmark evaluating perception models across varying camera configurations.

## 2 RELATED WORK

**Vision-based 3D Detection.** The development of camera-only 3D perception has gained great momentum recently. Early works such as FCOS3D (Wang et al., 2021), which extended the 2D FCOS detector (Tian et al., 2020) by adding 3D object regression branches, paved the way for improvements in depth estimation via probabilistic modeling (Wang et al., 2022a; Chen et al., 2022a). Later methods like DETR3D (Wang et al., 2022c), PETR (Liu et al., 2022), and Graph-DETR3D (Chen et al., 2022b) applied transformer-based architectures with learnable object queries in 3D space, drawing from the foundations of DETR (Zhu et al., 2021; Wang et al., 2022b), by-passing the limitations of perspective-based detection. Recent works utilize bird's-eye view (BEV)

for better 3D understanding. BEVDet (Huang et al., 2021) and M²BEV (Xie et al., 2022) effectively extended the Lift-Splat-Shoot (LSS) framework (Philion & Fidler, 2020) for 3D object detection. CaDDN (Reading et al., 2021) introduced explicit depth supervision in the BEV transformation to improve depth estimation. In addition, BEVFormer (Li et al., 2022), CVT (Zhou & Krähenbühl, 2022), and Ego3RT (Lu et al., 2022) explored multi-head attention mechanisms for view transformation, demonstrating further improvements in consistency. To further enhance accuracy, BEVDet4D (Huang & Huang, 2022), BEVFormer (Li et al., 2022), and PETRv2 (Liu et al., 2023a) leveraged temporal cues in multi-camera object detection, showing significant improvements over single-frame methods.

**Cross Domain Perception.** The cross-camera configurations problem proposed in this paper lies in the area of cross-domain perception. Domain generalization or adaptation is to enhance model performance on varying domains without re-training. For 2D perception, numerous cross-domain methods, such as feature distribution alignment and pseudo-labeling (Muandet et al., 2013; Li et al., 2018; Dou et al., 2019; Facil et al., 2019; Chen et al., 2018; Xu et al., 2020; He & Zhang, 2020; Zhao et al., 2020), have primarily addressed domain shifts caused by environmental factors like rain or low light. Recent 3D driving perception works (Hao et al., 2024; Peng et al., 2023) focus on transfering the models trained on clean environment or perfect sensor situations to corrupted sensor and noisy environments, leading to several benchmarks and methods. Cross camera configuration is a relatively new topic in this area. While some works (Wang et al., 2023b) find that the model's overfitting to camera parameters can lead to degrade performance because the models learn the fixed observation perspectives, the driving perception across camera parameters has seldom been systematically investigated.

**Sensor Configuration.** Sensor configurations has been proven important in the design of perception systems (Joshi & Boyd, 2008; Xu et al., 2022). Despite being relatively new in autonomous driving research (Liu et al., 2019), sensor placement has gained significant attention. For instance, Hu et al. (2022) and Li et al. (2024b) were the first to explore multi-LiDAR setups for improving 3D object detection, and Li et al. (2024a) studied how combining LiDAR and cameras impacts multimodal detection systems. Several other studies (Jin et al., 2022; Kim et al., 2023; Cai et al., 2023; Jiang et al., 2023b) focused on the strategic positioning of roadside LiDAR sensors for vehicle-to-everything (V2X) communication, shifting away from in-vehicle sensor placements. Although many efforts have aimed to refine sensor configurations for better performance, the challenge of adapting perception models to different sensor setups has been largely overlooked. Our research is the first to explore the generalization of driving perception models across diverse camera configurations.

## 3 UNIDRIVE

### 3.1 PROBLEM FORMULATION

In real-world multi-camera driving systems, perception models are typically trained on a specific camera configuration with fixed intrinsic and extrinsic parameters. However, the performance of these models often deteriorates when applied to new camera configurations, where the cameras may have different placements, orientations, or intrinsic properties.

**Perception Across Multi-camera Configurations.** Given a set of cameras $\mathcal{C} = \{C_1, C_2, \ldots, C_J\}$, each characterized by its intrinsic matrix $\mathbf{K}^{C_j} \in \mathbb{R}^{3 \times 3}$ and extrinsic matrix $\mathbf{E}^{C_j} \in \mathbb{R}^{4 \times 4}$, where $j \in \{1, 2, \ldots, J\}$ and $J$ is the number of cameras. The images captured by these cameras are denoted as $\mathbf{I}^{C_j} \in \mathbb{R}^{H^{C_j} \times W^{C_j} \times 3}$, where $H^{C_j}$ and $W^{C_j}$ are the height and width of the image $\mathbf{I}^{C_j}$. When deploying the model trained on $\{C_1, C_2, \ldots, C_J\}$ to a new set of cameras $\{C_1', C_2', \ldots, C_{J'}'\}$ with different camera numbers and intrinsic and extrinsic parameters, the model may no longer effectively understand the 3D scene due to the differences between the training and testing configurations.

**Universal Multi-camera Representation.** To address the transferability of learned models across camera configurations, we attempt to design a universal representation, which transforms images from different camera configurations to a unified space before input to the deep learning network. To achieve this, we propose a **Virtual Camera Projection** approach, which re-projects the views $\mathbf{I}^{C_j}$ from the original cameras $\mathcal{C} = \{C_1, C_2, \ldots, C_J\}$ into a unified set of virtual camera configurations $\mathcal{V} = \{V_1, V_2, ..., V_K\}$, where $K$ is the number of virtual cameras. The image is represented as $\mathbf{I}^{V_k} \in \mathbb{R}^{H^{V_k} \times W^{V_K} \times 3}$, where $H^{V_k}$ and $W^{V_K}$ are the image sizes, and $k \in \{1, 2, \ldots, K\}$ indexes the

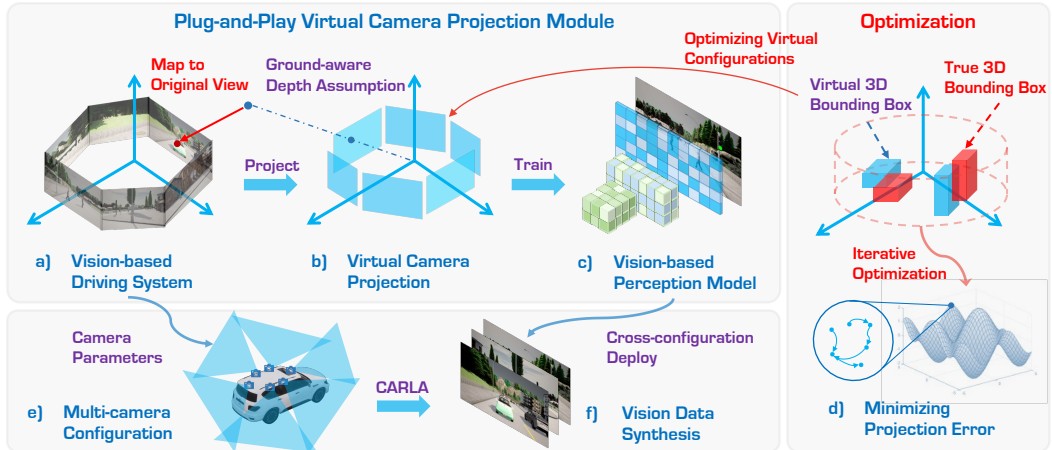

Figure 2: **Overview of UniDrive framework.** We transform the input images into a unified virtual camera space to achieve universal driving perception. To estimate the depth of pixels in the virtual view for projection, we propose a ground-aware depth assumption strategy. To obtain the most effective virtual camera space for multiple real camera configurations, we propose a data-driven CMA-ES (Hansen, 2016) based optimization strategy. To evaluate the efficacy of our framework, we propose an automatic data generation platform in CARLA (Dosovitskiy et al., 2017).

virtual camera views. We denote $\mathbf{K}^{V_k}$ and $\mathbf{E}^{V_k}$ as the intrinsic and extrinsic matrices for the virtual camera $V_k$. This virtual configuration serves as a standardized coordinate system for both training and inference, allowing the model to operate consistently across different physical camera setups.

## 3.2 VIRTUAL CAMERA PROJECTION

In this subsection, we explain the Virtual Camera Projection method to project points from multiple camera views onto virtual camera views using a combination of ground and cylindrical surface assumptions, as shown in Figure 2. The goal is to learn a transformation function $\mathcal{T}_{V \leftarrow C}$ that maps the images from the original cameras $\mathcal{C} = \{C_1, C_2, \ldots, C_J\}$ to the virtual cameras $\mathcal{V} = \{V_1, V_2, \ldots, V_K\}$ with minimum errors.

**Ground-aware Assumption.** For each pixel at coordinates $(u^{V_k}, v^{V_k})$ in the virtual view, its 3D coordinates in the virtual camera frame $(X_c^{V_k}, Y_c^{V_k}, Z_c^{V_k})$ are calculated based on the pixel's position in the image and the depth assumptions. Let the camera height be $h_c$, the focal lengths of the camera be $f_x^{V_k}$ and $f_y^{V_k}$, and the principal point (image center) be $(c_x^{V_k}, c_y^{V_k})$. We first project all pixels to the ground plane to compute the initial assumption of 3D coordinates in virtual camera frame as,

$$\left(\hat{X}_c^{V_k}, \hat{Y}_c^{V_k}, \hat{Z}_c^{V_k}\right) = \left(\frac{f_y^{V_k}\left(u^{V_k} - c_x^{V_k}\right)}{f_x^{V_k}\left(v^{V_k} - c_y^{V_k}\right)}\, h_c,\ h_c,\ \frac{f_y^{V_k}}{v^{V_k} - c_y^{V_k}}\, h_c\right). \tag{1}$$

The Euclidean distance to optical center is computed as $\hat{D}_c^{V_k} = \left\|\left(\hat{X}_c^{V_k}, \hat{Y}_c^{V_k}, \hat{Z}_c^{V_k}\right)\right\|_2$. Then we compare the distance $\hat{D}_c^{V_k}$ with threshold $D_0$, if $\hat{D}_c^{V_k} < D_0$, the points connected to corresponding pixels in the images are assumed on the ground, $(X_c^{V_k}, Y_c^{V_k}, Z_c^{V_k}) = (\hat{X}_c^{V_k}, \hat{Y}_c^{V_k}, \hat{Z}_c^{V_k})$. If $\hat{D}_c^{V_k} \geq D_0$, we assume that the points lie on a cylindrical-like surface at a fixed distance $D_0$ from the camera's optical center. In this case, the 3D coordinates are computed as:

$$\left(X_c^{V_k}, Y_c^{V_k}, Z_c^{V_k}\right) = \left(\frac{\left(u^{V_k} - c_x^{V_k}\right) D_0}{f_x^{V_k} d^{V_k}},\ \frac{\left(v^{V_k} - c_y^{V_k}\right) D_0}{f_y^{V_k} d^{V_k}},\ \frac{D_0}{d^{V_k}}\right), \tag{2}$$

where $d^{V_k} = \left\|\left(\frac{u^{V_k} - c_x^{V_k}}{f_x^{V_k}},\ \frac{v^{V_k} - c_y^{V_k}}{f_y^{V_k}},\ 1\right)\right\|_2$.

**Point-wise Projection.** Once the 3D coordinates $(X_c^{V_k}, Y_c^{V_k}, Z_c^{V_k})$ in the virtual camera frame are calculated, we transform the point into the world coordinate system with extrinsic matrix $\mathbf{E}^{V_k}$,

$\mathbf{p}_{\mathrm{w}} = \mathbf{E}^{V_k} \cdot \mathbf{p}_{\mathrm{c}}^{V_k}$, where $\mathbf{p}_{\mathrm{c}}^{V_k} = (X_{\mathrm{c}}^{V_k}, Y_{\mathrm{c}}^{V_k}, Z_{\mathrm{c}}^{V_k}, 1)^{\top}$ is the homogeneous coordinate of the point in the virtual camera's frame, and $\mathbf{p}_{\mathrm{w}} \in \mathbb{R}^4$ is the 3D point in the world coordinate system. Next, we transform the point from the world coordinate system into the original camera's coordinate system using the inverse of the original camera's extrinsic matrix $\mathbf{p}_{\mathrm{c}}^{C_j} = \mathbf{E}^{C_j\,-1} \cdot \mathbf{p}_{\mathrm{w}}$. Finally, we project the point back onto the original camera's 2D image plane using its intrinsic matrix, $(u^{C_j}, v^{C_j}, 1)^{\top} = \mathbf{K}^{C_j} \cdot \mathbf{p}_c^{C_j} = \mathbf{K}^{C_j} \cdot \mathbf{E}^{C_j\,-1} \cdot \mathbf{p}_{\mathrm{w}}$. This provides the pixel coordinates $(u^{C_j}, v^{C_j})$ in the original view that correspond to the pixel $(u^{V_k}, v^{V_k})$ in the virtual view. We denote $\mathbf{P}_{V_k \leftarrow C_j}(\hat{D}_{\mathrm{c}}^{V_k})$ as the projection transform matrix from $(u^{C_j}, v^{C_j})$ in the $i$-th original view to $(u^{V_k}, v^{V_k})$ based on the Euclidean distance to virtual camera optical center $\hat{D}_{\mathrm{c}}^{V_k}$.

**Image-level Transformation.** The point-wise projection is extended to the entire image view. For each pixel $(u^{V_k}, v^{V_k})$ in the $k$-th virtual view, we compute the corresponding pixel $(u^{C_j}, v^{C_j})$ in the $i$-th original view based on the projection matrix $\mathbf{P}_{V_k \leftarrow C_j}(\hat{D}_{\mathrm{c}}^{V_k})$. The entire image $\mathbf{I}^{C_j}$ of the $i$-th original view is warped into the virtual view $\mathbf{I}^{V_k \leftarrow C_j}$ as follows, $\mathbf{I}^{V_k \leftarrow C_j} = \mathcal{T}(\mathbf{I}^{C_j}, \mathbf{P}_{V_k \leftarrow C_j}(\hat{D}_{\mathrm{c}}^{V_k}))$, where $\mathcal{T}(\mathbf{I}, \mathbf{P})$ represents the warping function applied to the image $\mathbf{I}^{C_j}$ using the projection matrix $\mathbf{P}_{V_k \leftarrow C_j}(\hat{D}_{\mathrm{c}}^{V_k})$ based on the $\hat{D}_{\mathrm{c}}^{V_k}$.

**Blending Multiple Views.** Since each pixel in a single virtual view may have corresponding pixels from various original view, after transforming each original view into the virtual view, we merge

---

**Algorithm 1** Virtual Camera Projection

1: **Input:** $\{C_j, \mathbf{K}^{C_j}, \mathbf{E}^{C_j}, \mathbf{I}^{C_j}\}_{j=1}^{J}, \{V_k, \mathbf{K}^{V_k}, \mathbf{E}^{V_k}\}_{k=1}^{K}$
2: **Output:** $\{\mathbf{I}^{V_k}\}_{k=1}^{K}$
3: **for** $k = 1, 2, \ldots, K$ **do**
4:     **for** $(u^{V_k}, v^{V_k})$ in $\mathbf{I}^{V_k}$ **do**
5:         Compute $(\hat{X}_{\mathrm{c}}^{V_k}, \hat{Y}_{\mathrm{c}}^{V_k}, \hat{Z}_{\mathrm{c}}^{V_k}), \hat{D}_{\mathrm{c}}^{V_k}$ using equation 1
6:         **if** $\hat{D}_{\mathrm{c}}^{V_k} < D_0$ **then**
7:             $(X_{\mathrm{c}}^{V_k}, Y_{\mathrm{c}}^{V_k}, Z_{\mathrm{c}}^{V_k}) \leftarrow (\hat{X}_{\mathrm{c}}^{V_k}, \hat{Y}_{\mathrm{c}}^{V_k}, \hat{Z}_{\mathrm{c}}^{V_k})$
8:         **else**
9:             Compute $(X_{\mathrm{c}}^{V_k}, Y_{\mathrm{c}}^{V_k}, Z_{\mathrm{c}}^{V_k})$ using equation 2
10:         **end if**
11:         $\mathbf{p}_{\mathrm{w}} \leftarrow \mathbf{E}^{V_k} \cdot \mathbf{p}_{\mathrm{c}}^{V_k}, \mathbf{p}_{\mathrm{c}}^{V_k} = (X_{\mathrm{c}}^{V_k}, Y_{\mathrm{c}}^{V_k}, Z_{\mathrm{c}}^{V_k})$
12:         $\mathbf{p}_{\mathrm{c}}^{C_j} \leftarrow \mathbf{E}^{C_j\,-1} \cdot \mathbf{p}_{\mathrm{w}}$
13:         $(u^{C_j}, v^{C_j}) \leftarrow \mathbf{K}^{C_j} \cdot \mathbf{p}_{\mathrm{c}}^{C_j}$
14:         $\mathbf{I}^{V_k \leftarrow C_j}(u^{V_k}, v^{V_k}) \leftarrow \mathbf{I}^{C_j}(u^{C_j}, v^{C_j})$
15:     **end for**
16: **end for**
17: $\mathbf{I}^{V_k} \leftarrow \frac{1}{\mathbf{W}} \sum_{j=1}^{J} w_j \cdot \mathbf{I}^{V_k \leftarrow C_j}$ using equation 3

---

all the transformed images $\mathbf{I}^{V_k \leftarrow C_j}$ to form the final output image $\mathbf{I}^{V_k}$. This blending is performed by computing a weighted sum of all the projected views:

$$\mathbf{I}^{V_k} = \frac{1}{\mathbf{W}} \sum_{j=1}^{J} w_j \cdot \mathbf{I}^{V_k \leftarrow C_j} = \frac{1}{\mathbf{W}} \sum_{i=1}^{J} w_j \cdot \mathcal{T}(\mathbf{I}^{C_j}, \mathbf{P}_{V_k \leftarrow C_j}(\hat{D}_{\mathrm{c}}^{V_k})), \qquad (3)$$

where $\mathbf{W} = \sum_{j=1}^{J} w_j$ is the total weight, and $w_j$ is the blending weight for the $j$-th original view. The weights can be based on factors such as the angular distance between the original and virtual views, or the proximity of the cameras. We presented the detailed computation process in Algorithm 1.

### 3.3 VIRTUAL PROJECTION ERROR

To evaluate the accuracy of the Virtual Camera Projection method in the context of a 3D object detection task, we propose a weighted projection error metric based on angular discrepancies between the virtual and original camera views. This method accounts for both angular deviations and the distance from the camera's optical center to provide a more robust error evaluation.

**Angle Computation.** Given a driving scenario of 3D bounding box information, for each 3D bounding box $b_n = \{(x_{n,m}, y_{n,m}, z_{n,m})^{\top}\}_{m=1}^{8}$, we first project its corner points onto the original camera $C_j$ as the pixel $(u_{n,m}^{C_j}, v_{n,m}^{C_j})$, using the intrinsic matrix $\mathbf{K}^{C_j}$ and extrinsic matrix $\mathbf{E}^{C_j}$. Then, we use the inverse of the warping process $\mathbf{P}_{V_k \leftarrow C_j}$ to find the corresponding pixel $(u_{n,m}^{V_k}, v_{n,m}^{V_k})$ in the virtual camera view $V_k$ for each corner point. We compute the pitch angle $\theta_{n,m}^{V_k}$ and yaw angle $\phi_{n,m}^{V_k}$ relative to the virtual camera's optical center:

$$\left(\theta_{n,m}^{V_k}, \phi_{n,m}^{V_k}\right) = \left(\arctan \frac{v_{n,m}^{V_k} - c_y^{V_k}}{f_y^{V_k}}, \arctan \frac{u_{n,m}^{V_k} - c_x^{V_k}}{f_x^{V_k}}\right). \qquad (4)$$

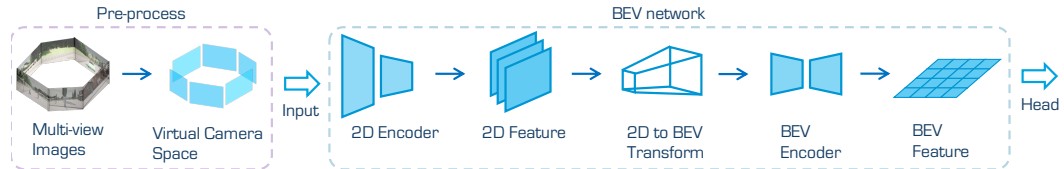

Figure 3: **Integration with Existing Methods.** Our virtual camera projection can be integrated into the pipeline as a pre-processing stage before feeding the multi-view images into the network.

---

**Algorithm 2** Virtual Camera Configuration Optimization

---

1: Initialize: $t \leftarrow 0$, $\mathbf{m}^{(0)}$, $\sigma^{(0)}$, $\mathbf{C}^{(0)}$, $N_t$, $M_t$, $\forall t \in \{0, 1, 2, \ldots, T\}$
2: **for** $t = 0, 1, 2, \ldots, T$ **do**
3:      **for** $i = 1$ to $N_t$ **do**
4:          Sample $\mathbf{u}_i^{(t)} \sim \mathcal{N}(\mathbf{m}^{(t)}, (\sigma^{(t)})^2 \mathbf{C}^{(t)})$ from $\delta$-density gird-level candidates
5:          Calculate $\mathcal{E}(\mathbf{u}_i^{(t)})$
6:      **end for**
7:      Update $\mathbf{m}^{(t+1)}$ based on the top $M_t$ best solutions $\hat{\mathbf{u}}_i^{(t)}$ via equation 7
8:      Update $\sigma^{(t+1)}$ and $\mathbf{C}^{(t+1)}$ via equation 8, equation 9, equation 10, and equation 11
9: **end for**

---

Next, for the same corner points, we directly project to the virtual view using $\mathbf{K}^{V_k}$ and $\mathbf{E}^{V_k}$ as $(u_{n,m}^{V_k\prime}, v_{n,m}^{V_k\prime})$. Then the pitch angle $\theta_{n,m}^{V_k\prime}$ and yaw angle $\phi_{n,m}^{V_k\prime}$ are

$$\left(\theta_{n,m}^{V_k\prime}, \phi_{n,m}^{V_k\prime}\right) = \left(\arctan \frac{v_{n,m}^{V_k\prime} - c_y^{V_k}}{f_y^{V_k}}, \arctan \frac{u_{n,m}^{V_k\prime} - c_x^{V_k}}{f_x^{V_k}}\right). \tag{5}$$

**Angle Error Calculation.** For each corner point, we compute the angular error between the original camera projection and the corresponding point in the virtual camera. The absolute errors in pitch and yaw are $\Delta\theta_{n,m}^{V_k} = \left|\theta_{n,m}^{V_k} - \theta_{n,m}^{V_k\prime}\right|$, $\Delta\phi_{n,m}^{V_k} = \left|\phi_{n,m}^{V_k} - \phi_{n,m}^{V_k\prime}\right|$. We use the distance $D_{n,m}^{V_k}$ of each corner point from the original camera's optical center as a weight. The distance is computed as $D_{n,m}^{V_k} = \left\|(x_{n,m}^{V_k}, y_{n,m}^{V_k}, z_{n,m}^{V_k})\right\|$. The weighted error for each corner point is then calculated as $\mathcal{E}_{n,m}^{V_k} = D_{n,m}^{V_k} \cdot (\Delta\theta_{n,m}^{V_k} + \Delta\phi_{n,m}^{V_k})$. The overall error for a 3D bounding box $b_n$ is obtained by summing the weighted errors of its eight corner points $\mathcal{E}_{b_n}^{V_k} = \sum_{m=1}^{8} \mathcal{E}_{n,m}^{V_k}$. We sum the projection errors across all 3D bounding boxes $b_n \in \mathcal{B}$ to compute the total projection error

$$\mathcal{E} = \sum_{n=1}^{N} \mathcal{E}_{b_n}^{V_k} = \sum_{n=1}^{N} \sum_{m=1}^{8} \mathcal{E}_{n,m}^{V_k}. \tag{6}$$

## 3.4 Optimizing Virtual Camera Configurations

Given a set of multi-camera systems, we aim to design a unified virtual camera configuration that minimizes the reprojection error across all original camera configurations. To achieve this, we adopt the heuristic optimization based on the Covariance Matrix Adaptation Evolution Strategy (CMA-ES) (Hansen, 2016) to find an optimized set of virtual camera configurations.

**Objective Function.** Given multiple driving perception systems with varying multi-camera confgirations indexed by $s$, the total error across all systems is expressed as $\mathcal{E}_{total} = \sum_{s=1}^{S} \mathcal{E}^{(s)}(\mathbf{u})$, where $\mathbf{u} = \{V_k, \mathbf{K}^{V_k}, \mathbf{E}^{V_k}\}_{k=1}^{K}$ includes both the intrinsic and extrinsic camera parameters of virtual multi-camera framework, $K$ is the total quantity of virtual cameras and $S$ is the total quantity of multi-camera driving systems that share the same perception model. We aim to minimize this error by sampling and updating the virtual camera parameters iteratively through a CMA-ES based optimization method.

**Optimization Method.** Our Optimization strategy begins by defining a multivariate normal distribution $\mathcal{N}(\mathbf{m}^{(t)}, (\sigma^{(t)})^2 \mathbf{C}^{(t)})$, where $\mathbf{m}^{(t)}$ represents the mean vector, $\sigma^{(t)}$ denotes the step size, and $\mathbf{C}^{(t)}$ is the covariance matrix at iteration $t$. The configuration space $\mathcal{U}$ is discretized with a density

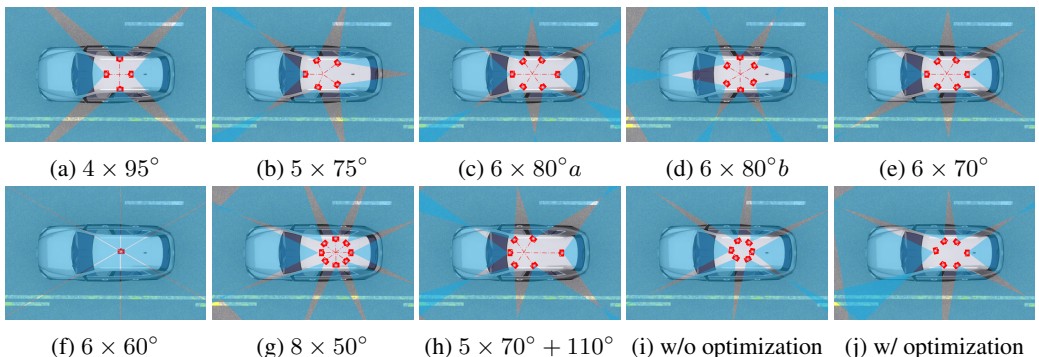

(a) $4 \times 95°$   (b) $5 \times 75°$   (c) $6 \times 80°a$   (d) $6 \times 80°b$   (e) $6 \times 70°$

(f) $6 \times 60°$   (g) $8 \times 50°$   (h) $5 \times 70° + 110°$   (i) w/o optimization   (j) w/ optimization

Figure 4: **Visualized multi-camera configurations.** We illustrate the multi-view camera configurations used in our study. There configurations are inspired by practical applications in the industry.

$\delta$, and $N_t$ candidate configurations $\mathbf{u}_i^{(t)} \sim \mathcal{N}(\mathbf{m}^{(t)}, (\sigma^{(t)})^2 \mathbf{C}^{(t)})$ are sampled at each iteration $t$. Initialization begins with the initial mean $\mathbf{m}^{(0)}$, step size $\sigma^{(0)}$, and covariance matrix $\mathbf{C}^{(0)} = \mathbf{I}$. The updated mean vector $\mathbf{m}^{(t+1)}$ is calculated in the subsequent iteration to serve as the new center for the search distribution concerning the virtual camera configuration. The process can be mathematically expressed as:

$$\mathbf{m}^{(t+1)} = \sum_{i=1}^{M_t} w_i \hat{\mathbf{u}}_i^{(t)}, \ \mathcal{E}(\hat{\mathbf{u}}_1^{(t)}) \geq \mathcal{E}(\hat{\mathbf{u}}_2^{(t)}) \geq \cdots \geq \mathcal{E}(\hat{\mathbf{u}}_{M_t}^{(t)}), \tag{7}$$

where $M_t$ is the number of top solutions selected to update $\mathbf{m}^{(t+1)}$, and $w_i$ are weights determined by solution performance. The evolution path $\mathbf{p}_{\mathbf{C}}^{(t+1)}$, which tracks the direction of successful optimization steps, is updated as:

$$\mathbf{p}_{\mathbf{C}}^{(t+1)} = (1 - c_{\mathbf{C}}) \cdot \mathbf{p}_{\mathbf{C}}^{(t)} + \sqrt{1 - (1 - c_{\mathbf{C}})^2} \cdot \sqrt{\frac{1}{\sum_{i=1}^{M_t} w_i^2}} \cdot \frac{\mathbf{m}^{(t+1)} - \mathbf{m}^{(t)}}{\sigma^{(t)}}, \tag{8}$$

where $c_{\mathbf{C}}$ is the learning rate for updating the covariance matrix. The covariance matrix $\mathbf{C}$, which defines the distribution's shape for camera configurations, is adjusted at each iteration as follows:

$$\mathbf{C}^{(t+1)} = (1 - c_{\mathbf{C}})\mathbf{C}^{(t)} + c_{\mathbf{C}} \mathbf{p}_{\mathbf{C}}^{(t+1)} \mathbf{p}_{\mathbf{C}}^{(t+1)^T}. \tag{9}$$

Similarly, the evolution path for the step size, $\mathbf{p}_\sigma$, is updated, and the global step size $\sigma$ is then adjusted to balance exploration and exploitation:

$$\mathbf{p}_\sigma^{(t+1)} = (1 - c_\sigma)\mathbf{p}_\sigma^{(t)} + \sqrt{1 - (1 - c_\sigma)^2} \cdot \sqrt{\frac{1}{\sum_{i=1}^{M_t} w_i^2}} \cdot \frac{\mathbf{m}^{(t+1)} - \mathbf{m}^{(t)}}{\sigma^{(t)}}, \tag{10}$$

$$\sigma^{(t+1)} = \sigma^{(t)} \exp\left(\frac{c_\sigma}{d_\sigma}\left(\frac{\|\mathbf{p}_\sigma^{(t+1)}\|}{E\|\mathcal{N}(0, \mathbf{I})\|} - 1\right)\right), \tag{11}$$

where $c_\sigma$ is the learning rate for updating $\mathbf{p}_\sigma$, and $d_\sigma$ is a normalization factor controlling the adjustment rate of the global step size. We presented the detailed optimization process in Algorithm 2.

# 4 EXPERIMENTS

## 4.1 BENCHMARK SETUPS

**Data Generation.** We generate multi-view image data and 3D objects ground truth in CARLA simulator (Dosovitskiy et al., 2017). We use the maps of Towns 1-6 to collect data. We incorporate 6 classes for 3D object detection, including *Car*, *Bus*, *Truck*, *Motorcycle*, *Bicycle*, and *Pedestrian*. The dataset consists of 500 scenes (20,000 frames) for each camera configuration. We split 250 scenes

Table 1: **Quantitative results of BEVFusion-C for 3D detection across camera configurations**. The detector is trained on the blue configurations and tested on all configurations directly. We report the mAP (↑) and class-level AP (↑) scores in percentage (%).

| Configurations | mAP | Car | Bus | Truck | Ped. | Motor | Bic. | Configurations | mAP | Car | Bus | Truck | Ped. | Motor | Bic. |
|---|---|---|---|---|---|---|---|---|---|---|---|---|---|---|---|
| $5 \times 70° + 110°$ | 63.9 | 62.4 | 58.0 | 66.5 | 54.7 | 68.7 | 72.9 | $6 \times 60$ | 69.3 | 68.7 | 67.4 | 66.3 | 62.2 | 78.4 | 72.8 |
| $4 \times 95°$ | 4.9 | 4.6 | 5.1 | 3.8 | 3.9 | 3.1 | 4.2 | $4 \times 95°$ | 3.8 | 4.1 | 4.3 | 3.2 | 4.1 | 3.3 | 3.6 |
| $5 \times 75°$ | 7.2 | 9.5 | 4.8 | 6.2 | 5.1 | 9.0 | 8.3 | $5 \times 75°$ | 3.4 | 3.7 | 2.5 | 3.1 | 2.7 | 4.3 | 4.2 |
| $6 \times 80°a$ | 8.5 | 11.7 | 8.8 | 8.4 | 6.1 | 8.2 | 7.7 | $6 \times 80°a$ | 0.6 | 1.7 | 0.4 | 0.5 | 0.1 | 0.4 | 0.6 |
| $6 \times 80°b$ | 6.9 | 10.0 | 7.2 | 7.8 | 5.2 | 6.1 | 5.6 | $6 \times 80°b$ | 0.4 | 1.4 | 0.0 | 0.7 | 0.0 | 0.2 | 0.1 |
| $6 \times 70°$ | 67.5 | 65.2 | 61.2 | 69.3 | 57.9 | 79.5 | 72.1 | $6 \times 70°$ | 8.1 | 9.6 | 4.3 | 6.8 | 7.1 | 11.0 | 10.0 |
| $6 \times 60°$ | 9.2 | 12.4 | 7.0 | 8.0 | 6.5 | 11.9 | 9.4 | $5 \times 70° + 110°$ | 4.6 | 4.9 | 3.0 | 3.5 | 3.4 | 5.4 | 7.4 |
| $8 \times 50°$ | 0.5 | 0.6 | 0.1 | 0.9 | 0.2 | 0.3 | 0.6 | $8 \times 50°$ | 17.3 | 18.5 | 9.9 | 14.1 | 16.7 | 21.2 | 23.4 |
| $6 \times 80°a$ | 66.7 | 65.4 | 66.2 | 63.7 | 55.8 | 75.9 | 72.9 | $6 \times 80°b$ | 69.1 | 66.0 | 65.1 | 72.1 | 58.3 | 78.6 | 74.2 |
| $4 \times 95°$ | 3.8 | 4.3 | 5.0 | 3.6 | 3.2 | 2.8 | 3.9 | $4 \times 95°$ | 3.5 | 3.9 | 4.1 | 3.3 | 3.2 | 2.6 | 3.7 |
| $5 \times 75°$ | 30.4 | 31.2 | 23.4 | 27.8 | 28.6 | 36.9 | 34.2 | $5 \times 75°$ | 29.6 | 30.3 | 22.6 | 27.1 | 27.9 | 36.3 | 33.2 |
| $5 \times 70° + 110°$ | 9.2 | 10.5 | 6.5 | 8.6 | 7.1 | 8.8 | 13.3 | $6 \times 80°a$ | 63.2 | 65.5 | 67.0 | 66.4 | 46.7 | 68.2 | 65.1 |
| $6 \times 80°b$ | 63.3 | 65.4 | 63.8 | 70.9 | 46.3 | 68.1 | 65.4 | $6 \times 60°$ | 1.7 | 2.9 | 0.5 | 1.1 | 0.7 | 2.3 | 2.6 |
| $6 \times 70°$ | 16.4 | 18.0 | 9.4 | 13.3 | 14.7 | 22.2 | 20.6 | $6 \times 70°$ | 16.1 | 17.7 | 8.3 | 12.3 | 14.6 | 23.7 | 19.9 |
| $6 \times 60°$ | 1.8 | 3.3 | 0.8 | 1.5 | 0.6 | 2.3 | 2.4 | $5 \times 70° + 110°$ | 8.9 | 10.3 | 5.6 | 7.5 | 7.1 | 9.6 | 13.4 |
| $8 \times 50°$ | 0.4 | 0.5 | 0.0 | 0.8 | 0.1 | 0.1 | 0.6 | $8 \times 50°$ | 0.2 | 0.4 | 0.0 | 0.9 | 0.3 | 0.3 | 0.4 |

for training and 250 scenes for validation. Our dataset is organized as the format of nuScenes (Caesar et al., 2020) and compatible to the `nuscenes-devkit` python package for convenient processing.

**Camera Configurations.** We adopt several commonly used camera configurations in automotive practice with various camera quantities, placements and field of views. These configurations are represented in Figure 4. We set all camera resolutions to `1600×900` as nuScenes. Our camera configurations include camera numbers from 4 to 8. For the field of view (FOV) for cameras, we conduct study mainly on 6 cameras with FOV = 60, 70, 80. For the placement, we design two different types of placement as shown in Figure 4 (c), (d). We also include the original configurations of nuScenes (Caesar et al., 2020) dataset with five 70° cameras and a 110° camera.

**Deteciton Method.** Due to the extensive computation resource needed to benchmark the multi-camera configurations, we only compare our method with the camera variant of BEVFusion (Liu et al., 2023b) (abbreviated as BEVFusion-C). BEVFusion is one of the representative methods in many leaderboards, such as nuScenes (Caesar et al., 2020) and Waymo (Sun et al., 2020).

## 4.2 COMPARATIVE STUDY

We conduct comparative studies to evaluate the performance of camera perception across configurations. Through our analysis, we are able to demonstrate the effectiveness of UniDrive framework.

**Effectiveness of UniDrive.** In Table 1 and 2, we present the 3D object detection results of BEVFusion-C (Liu et al., 2023b) and UniDrive. The models are trained on one configuration and tested on other varying camera configurations. The performance of BEVFusion-C degrades a lot when deployed on cross-camera configuration tasks, nearly unusable on other configurations. As shown in Table 2, we train the models using our plug-and-play UniDrive framework. The detection performance significantly improves compared to BEVFusion-C (Liu et al., 2023b). Our method only experiences little performance degradation on cross-camera configuration tasks. We present more results in Figure 5, which comprehensively shows the effectiveness of our framework.

**Optimization via UniDrive.** To demonstrate the importance of optimization in UniDrive, we compare the perception performance between optimized virtual camera configurations and intuitive one in Figure 5. The intuitive virtual camera configuration places all cameras in the center of the vehicle roof. As shown in Figure 5 (b), although the intuitive setup (without optimizing) also significantly improved cross-camera configuration perception performance compared to BEVFusion-C (in Figure 5 (a)), it exhibited a clear preference for certain configurations while performing poorly on others. In contrast, the optimized virtual camera parameters (in Figure 5 (c)) demonstrated greater

Table 2: **Quantitative results of UniDrive for 3D detection across camera configurations**. The detector is trained on the blue configurations and tested on all configurations directly. We report the mAP (↑) and class-level AP (↑) scores in percentage (%).

| Configurations | mAP | Car | Bus | Truck | Ped. | Motor | Bic. | Configurations | mAP | Car | Bus | Truck | Ped. | Motor | Bic. |
|---|---|---|---|---|---|---|---|---|---|---|---|---|---|---|---|
| $5 \times 70° + 110°$ | 68.8 | 67.5 | 64.8 | 71.9 | 59.1 | 73.6 | 75.9 | $6 \times 60°$ | 64.6 | 63.4 | 58.2 | 59.7 | 59.2 | 76.7 | 70.0 |
| $4 \times 95°$ | 60.1 | 59.1 | 57.2 | 58.4 | 59.2 | 68.6 | 67.8 | $4 \times 95°$ | 58.9 | 57.7 | 54.1 | 56.9 | 53.1 | 65.2 | 67.4 |
| $5 \times 75°$ | 66.7 | 64.5 | 65.9 | 67.8 | 59.6 | 72.3 | 70.1 | $5 \times 75°$ | 62.2 | 59.5 | 60.6 | 65.8 | 53.6 | 70.3 | 63.1 |
| $6 \times 80°a$ | 69.4 | 68.4 | 68.1 | 67.5 | 57.8 | 78.2 | 76.1 | $6 \times 80°a$ | 64.4 | 65.1 | 64.9 | 65.5 | 53.3 | 70.2 | 67.1 |
| $6 \times 80°b$ | 65.8 | 64.7 | 62.0 | 63.9 | 55.8 | 76.7 | 71.7 | $6 \times 80°b$ | 65.7 | 63.2 | 63.1 | 63.9 | 55.8 | 76.7 | 71.7 |
| $6 \times 70°$ | 68.4 | 66.8 | 64.3 | 69.8 | 57.6 | 79.1 | 72.8 | $6 \times 70°$ | 65.0 | 62.9 | 60.8 | 65.4 | 55.1 | 73.0 | 72.8 |
| $6 \times 60°$ | 63.1 | 60.6 | 57.4 | 58.8 | 59.2 | 73.0 | 69.6 | $5 \times 70° + 110°$ | 63.6 | 61.2 | 61.4 | 64.8 | 55.2 | 71.0 | 68.1 |
| $8 \times 50°$ | 58.9 | 57.1 | 55.4 | 56.1 | 51.1 | 69.7 | 64.1 | $8 \times 50°$ | 63.8 | 60.2 | 58.3 | 62.6 | 56.8 | 74.2 | 70.5 |
| $6 \times 80°a$ | 69.4 | 69.0 | 67.7 | 66.6 | 58.6 | 78.4 | 76.1 | $6 \times 80°b$ | 63.1 | 63.2 | 61.9 | 59.8 | 53.4 | 71.1 | 69.4 |
| $4 \times 95°$ | 55.9 | 56.4 | 58.4 | 52.7 | 48.3 | 59.5 | 60.1 | $4 \times 95°$ | 53.6 | 51.2 | 48.0 | 49.7 | 51.0 | 61.5 | 60.1 |
| $5 \times 75°$ | 65.2 | 63.6 | 64.8 | 66.8 | 57.0 | 70.9 | 68.3 | $5 \times 75°$ | 62.7 | 62.1 | 60.6 | 62.2 | 56.3 | 69.9 | 67.0 |
| $5 \times 70° + 110°$ | 63.7 | 61.3 | 58.9 | 60.5 | 59.9 | 72.8 | 68.8 | $6 \times 80°a$ | 64.5 | 62.6 | 60.3 | 62.4 | 58.6 | 71.4 | 71.6 |
| $6 \times 80°b$ | 66.2 | 65.2 | 61.1 | 65.1 | 56.1 | 76.3 | 73.2 | $6 \times 60°$ | 62.6 | 60.3 | 59.1 | 62.6 | 53.4 | 70.2 | 70.2 |
| $6 \times 70°$ | 68.9 | 67.9 | 63.5 | 70.7 | 58.3 | 79.5 | 73.8 | $6 \times 70°$ | 62.5 | 59.4 | 55.6 | 62.9 | 52.6 | 73.4 | 70.8 |
| $6 \times 60°$ | 59.6 | 57.2 | 54.0 | 55.7 | 57.0 | 67.5 | 66.1 | $5 \times 70° + 110°$ | 57.9 | 56.1 | 52.5 | 53.7 | 56.9 | 64.3 | 63.7 |
| $8 \times 50°$ | 61.2 | 60.3 | 58.1 | 59.9 | 54.5 | 68.2 | 65.9 | $8 \times 50°$ | 58.4 | 60.3 | 57.0 | 54.3 | 53.9 | 64.4 | 60.2 |

adaptability, showing relatively consistent performance across various configurations. This is crucial for the concurrent development of multiple multi-camera perception systems in autonomous driving.

## 4.3 ABLATION STUDY

In this section, we analyze the interplay between our proposed virtual projection strategy and perception performance to address these questions: *1) What's the impact of camera extrinsic and intrinsic for cross-configuration perception? 2) How UniDrive works towards these parameters separately?*

**Camera Intrinsics.** Changes in camera intrinsics pose the greatest challenge for cross-camera parameter perception. In Figure 5 (a), BEVFusion-C almost entirely fails when tasked on distinct camera intrinsics with the detection accuracy mostly under *20%*. For instance, BEVFusion only gets *1.8%* when deploying models trained on $6 \times 80°a$ to $6 \times 60°$. In contrast, in Figure 6, our UniDrive framework demonstrates substantial robustness, with performance dropping by at most *9.8%* under the largest intrinsic differences, which highlights the effectiveness of our approach.

**Camera Height.** The variation in the vertical position of cameras can significantly impact perception performance, as cameras at varying heights capture images with distinct geometric features. We perform experiments specifically for varying camera heights at 1.6 meters, 1.4 meters, 1.8 meters, and 2.5 meters. We train the model on 1.6 meters and test on other configuraitons. As shown in Fig. 6b. BEVFusion-C experiences a substantial performance drop for more than *10%*, when faced with varying camera heights. In contrast, UniDrive significantly improves performance across different camera heights, demonstrating enhanced robustness with only *3.0%* performance decreasing.

**Camera Placement.** Changing the camera's horizontal position and orientation on presents a relatively smaller challenge for cross-camera parameter perception. As shown in Figure 5 (a), BEVFusion-C experiences a performance drop of *5.9%* when deploying the model trained on the $6 \times 80°b$ configuration to the $6 \times 80°a$ configuration. Nonetheless, our UniDrive framework further enhances cross-camera parameter perception performance. In Figure 5 (c), we train the model on the $6 \times 80°a$ configuration and test on other configurations, UniDrive only experiences a *4.6%* when deploying the model trained on the $6 \times 80°b$ configuration to the $6 \times 80°a$ configuration.

## 4.4 ANALYSIS

In this section, we further investigate some useful insight points found in the benchmark experiments: *1) What's the impact of inconsistency in multi-camera intrinsics for perception? 2) How UniDrive works towards this inconsistency?*

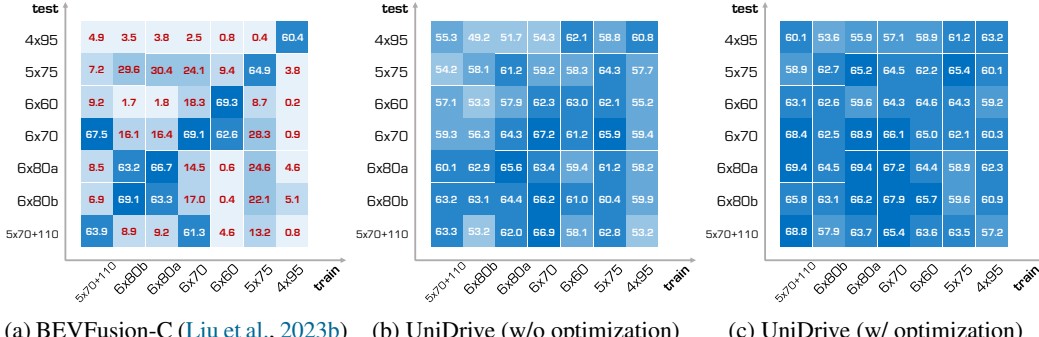

Figure 5: **Performance evaluations of BEVFusion-C and UniDrive** on 3D object detection across camera configurations. We report the mAP ($\uparrow$) scores in percentage ($\%$).

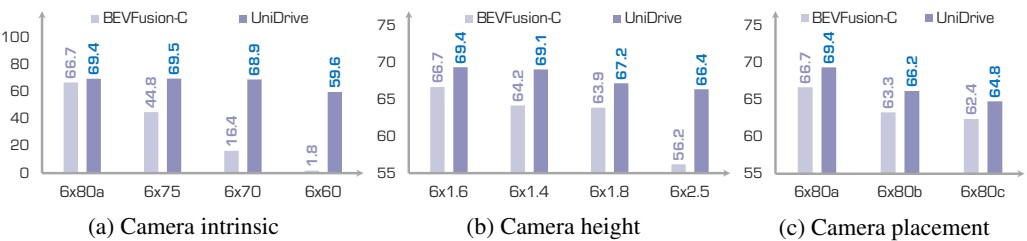

Figure 6: **Ablation Study of BEVFusion-C and UniDrive** on 3D object detection across camera configurations. We report the mAP ($\uparrow$) scores in percentage ($\%$).

**Degradation with Inconsistent Intrinsics.** In our experiments, we observed that for multi-camera systems, models perform better when camera intrinsics are consistent compared to when they vary. However, due to design aesthetics and other constraints, many autonomous driving companies use multiple cameras with different intrinsic parameters to achieve 360-degree perception. For instance, the nuScenes (Caesar et al., 2020) uses five $70°$ cameras and one $110°$ camera. As shown in Fig 1, BEVFusion-C performs a lot better in $6 \times 80°a$ and $6 \times 60°$ compared to configuration $5 \times 70° + 110°$. Thus, inconsistency in camera intrinsics can potentially hinder perception improvement.

**Improvement via UniDrive.** Our framework significantly enhances the perception performance of multi-camera systems with varying intrinsics by leveraging a virtual camera system with consistent intrinsics. For training and testing on the same configurations, as demonstrated in Figure 2, UniDrvie achieves *68.8%* accuracy in $5 \times 70° + 110°$ configuration, which surpasses *4.9%* than BEVFusion-C (*63.9%*). For testing across camera configurations, UniDrive experiences little accuracy reduction only in rare situations. This demonstrates that UniDrive has substantial potential to push advancements in driving perception technology.

## 5  CONCLUSION

In this paper, we introduce the UniDrive framework, a robust solution for enhancing the generalization of vision-centric autonomous driving models across varying camera configurations. By leveraging a unified set of virtual cameras and a ground-aware projection method, our approach effectively mitigates the challenges posed by camera intrinsics and extrinsics. The proposed virtual configuration optimization ensures minimal projection error, enabling adaptable and reliable performance across diverse sensor setups. Extensive experiments in CARLA validate the effectiveness of UniDrive, demonstrating strong generalization capabilities with minimal performance loss. Our framework not only serves as a plug-and-play module for existing 3D perception models but also paves the way for more versatile and scalable autonomous driving solutions.

**Limitation.** The camera configurations analyzed in this paper can not cover all real-world setups, more comprehensive experiments may be required. In addition, our research are fully conducted on simulation data, as real-world experiments are time-consuming and need extensive resource.

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

## A   VISUALIZATION

We present the visualization results of the virtual camera projection in Figure 7. Overall, the warping from the original view to the virtual view is highly accurate. Only a few areas are not warped because the original cameras lack coverage of those regions.

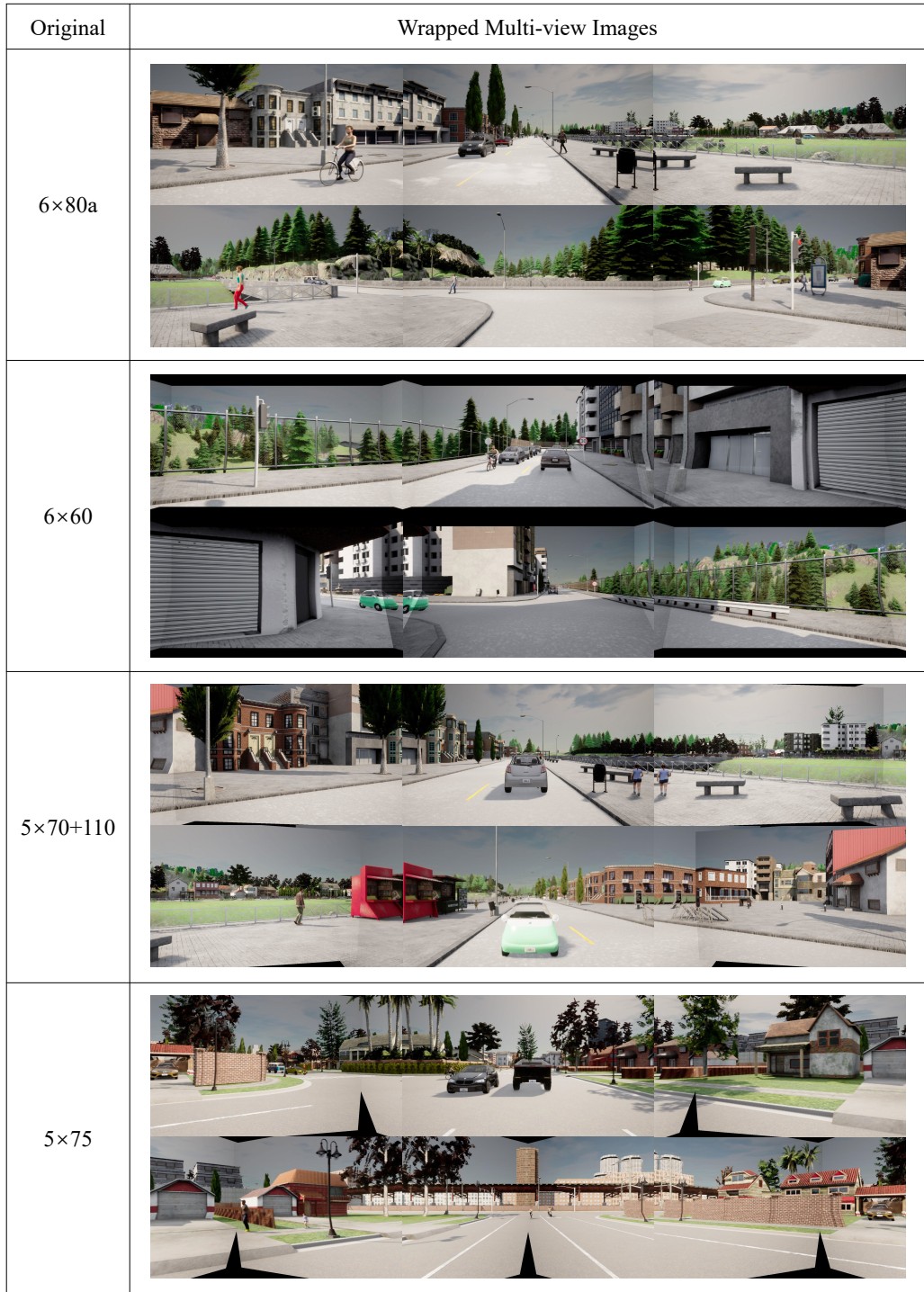

Figure 7: Wrapped Multi-view Images.

