# OpenReview forum: "UniDrive: Towards Universal Driving Perception Across Camera Configurations"
_ICLR.cc/2025/Conference — ICLR 2025 Poster_

### Official Review · Reviewer_DD4T · 2024-10-27

**Soundness:** 3
**Presentation:** 3
**Contribution:** 3
**Rating:** 6
**Confidence:** 4

**Summary:**

Camera-only 3D object detection is sensitive to camera settings. It has been proved that a detection model trained on a camera setting fails to provide satisfactory performance in another camera setting. The authors propose a method, UniDrive, to unify this. The proposed UniDrive comprises a set of virtual cameras, which wrap real cameras into virtual space and thus bridge the gap between different camera settings. It is worth noticing that the number of real and virtual cameras does not need to be the same. The authors also proposed a virtual projection error, which enables an optimization process to find the optimal virtual camera setting. Experiments are conveyed on the CARLA simulator and the BEVFusion-C model. It is shown that, with the proposed method, model performance on various camera settings is significantly improved.

**Strengths:**

- The article is clear and easy to follow.
- The motivation of the article is a long-standing problem in the 3D detection domain, and the authors propose a feasible way for solving the problem.
- Mathematical details are given for the projection from virtual cameras to real cameras.
- In the experiments, the proposed method is testified on seven different camera settings, showing its generalization to different camera settings.

**Weaknesses:**

- Only BEVFusion-C is used in the experiments. In line 407, the authors mentioned that BEVFusion-C is one of the SOTA methods in popular public leaderboards, which is not currently the case. The effectiveness of the proposed method should show its universal capability on multiple methods with at least 2-3 different camera settings, such as some methods spercifically designed camera-only detection like BEVFormer [1] and Sparse4D [2].
- It is not spercifically described that how to adopt the proposed method to exsiting network. It would be helpful as the proposed method serves as a plug-and-play component.


[1] Li, Zhiqi, et al. "Bevformer: Learning bird’s-eye-view representation from multi-camera images via spatiotemporal transformers." European conference on computer vision. Cham: Springer Nature Switzerland, 2022.

[2] Sun, Wenchao, et al. "SparseDrive: End-to-End Autonomous Driving via Sparse Scene Representation." arXiv preprint arXiv:2405.19620 (2024).

**Questions:**

- As mentioned in weaknesses, the effectiveness of the proposed method should be testified on multiple camera-only 3D detection methods.
- It is mentioned in line 302 that the virtual camera configuration is obtained through an optimization process based on a given set of camera systems; it is curious to know if the proposal method also works toward a randomly picked virtual camera setting. This would testify to the effectiveness of the proposed metric and optimization process in Sections 3.3 and 3.4. It is recommanded to compare the optimized virtual camera configuration to randomly selected configurations and evaluate performance across different camera settings.

---

> ### Author Response · Authors · 2024-11-23
> **Response to Reviewer DD4T**
>
> **Dear Reviewer `DD4T`,**
>
> Thanks for devoting time to this review and providing valuable comments. Below, we provide detailed responses to your comments and concerns. The revised manuscript has been uploaded, with all changes clearly highlighted in red for your convenience.
>
> ---
> > **W1 & Q1:** Results with more camera-only 3D detection methods.
>
> **A:** Thanks for the constructive suggestion! We agree that evaluating our framework on additional methods would further demonstrate its generalizability. We previously prioritized BEVFusion-C as our baseline method as it introduced an efficient BEV pooling operator and runs faster than the other methods, making it widely adopted in real-world applications. Due to the limited time in the discussion, we have conducted experiments with BEVFormer as shown below. UniDrive demonstrated promising adaptability, we will extend to the other models including Sparse4D.
>
> | | 6x60 (train) | 6x80a | 6x80b | 6x70 | 5x70 + 110 |
> |---|:-:|:-:|:-:|:-:|:-:|
> | BEVFormer | 71.8 | 4.2| 6.3 |29.6 |12.5 |
> | Ours | 69.1 | 66.5 | 69.4 | 67.3 | 63.1|
>
> | | 6x70 (train) | 6x80a | 6x80b | 6x60 | 5x70 + 110 |
> |---|:-:|:-:|:-:|:-:|:-:|
> | BEVFormer | 72.1 | 16.4 | 21.9 | 8.5 | 60.5
> | Ours | 68.7 | 64.9 | 63.7 | 65.2 | 63.9 |
>
> > **W2:** It is not specifically described that how to adopt the proposed method to existing network. It would be helpful as the proposed method serves as a plug-and-play component.
>
> **A:** Thank you for the valuable comment! We **add** a pipeline (as shown in **Figure 3**) in the revised paper and show how to use our module. Our module operates at the data input stage of the model as a preprocessing step.
>
> >**Q2:** It is recommended to compare the optimized virtual camera configuration to randomly selected configurations and evaluate performance across different camera settings.
>
> **A:** Thanks for your insightful question! As shown in **Figure 5** of the paper, we conducted experiments comparing the virtual camera configurations **with** optimization (*w/ optimization*) to randomly picked configurations **without** optimization (*w/o optimization*). The results demonstrate that the average transfer performance is significantly improved after optimization. This highlights the effectiveness of our proposed metric and optimization process in Sections 3.3 and 3.4.
>
> ---
>
> We sincerely thank you again for the valuable comments provided during this review.

---

> > ### Comment · Reviewer_DD4T · 2024-11-26
> >
> > Thank you for your response and experiments. There is still room to improve the paper. But I think the current version is valuable for the ICLR community. I will keep my rating.

---

> ### Author Response · Authors · 2024-12-01
>
> **Dear Reviewer `DD4T`,**
>
> Thank you for your positive recognition! We sincerely appreciate your constructive comments and will continue to refine our paper. Thank you once again for your time and effort during this review!

---

### Official Review · Reviewer_3zcR · 2024-11-02

**Soundness:** 4
**Presentation:** 3
**Contribution:** 3
**Rating:** 8
**Confidence:** 5

**Summary:**

I greatly appreciate this paper; in fact, I had similar ideas that I hadn't implemented. Seeing the author perfectly execute this idea and achieve excellent results is truly exciting. The paper proposes transforming images into a unified virtual camera space with a virtual configuration optimization strategy, thereby enhancing robustness to different camera configurations.

**Strengths:**

1. The paper proposes to transform images into a unified virtual camera space, improving robustness across various camera configurations.
2. Additionally, the paper proposes a virtual configuration optimization strategy that minimizes projection errors.
3. The paper also presents a systematic data generation platform and a benchmark for evaluating perception models under different camera configurations.

**Weaknesses:**

See the Questions section.

**Questions:**

1. Does the virtual configuration optimization strategy actually work effectively? （Core concern: I am highly willing to increase the score if this issue is explained thoroughly.）

   1.1 In certain camera configurations, the results from a single dataset in the paper are not as good as those of the original BEVFusion-C (e.g., 6x60 69.3 vs. 64.6, 6x80b 69.1 vs. 63.1). Why can't the virtual configuration optimization improve to achieve a virtual camera space as good as or better than the original BEVFusion-C?

   1.2 In some camera configurations, the results of individual optimization are not as good as those from transfer learning, such as 60x80b (train on 60x80b 63.1 vs. train on 60x70 67.9). Why can't the virtual configuration optimization achieve a virtual camera space as good as or better than the transfer setting?

   1.3 For different camera configurations, what do the optimized Virtual Cameras look like? Can you provide the optimized intrinsics and extrinsics?

   1.4 How well does the original view warp to the virtual view? Are there any areas that are not properly warped? Can you provide visual results?

2. While BEVFusion is popular, the paper focuses on vision-centric autonomous driving. It would be beneficial to include experiments with popular detectors in the camera domain, such as BEVFormer and PETR. These differ from methods like BEVFusion-C and BEVDet, as they are based on query-based transformer solutions.

3. Were the number of virtual cameras optimized together? If not, could you provide ablation experiments?

4. The paper discusses that Inconsistent Intrinsics have a significant impact. Since real autonomous vehicles also have inconsistent heights, could you include an experimental analysis?

   4.1 Were the heights of virtual cameras optimized together?

5. Given the various camera configurations, would training on datasets that mix various configurations to optimize a unified virtual camera space lead to improved results?

6. All experiments are conducted on simulation data. Incorporating transfer experiments with real datasets would be more convincing, like the real dataset transfer experiments conducted in 3D Domain Generalization/Adaptation.

Note: I have provided several questions. The authors can prioritize addressing the core issues based on their time constraints, and it is not necessary to conduct all experiments.

---

> ### Author Response · Authors · 2024-11-24
> **Response to Reviewer 3zcR (Part 1/2)**
>
> **Dear Reviewer `3zcR`,**
>
> Thanks for devoting time to this review and providing valuable comments. Below, we provide detailed responses to your comments and concerns. The revised manuscript has been uploaded, with all changes clearly highlighted in red for your convenience.
>
> ---
>
> >**Q1:** Does the virtual configuration optimization strategy actually work effectively? (Core concern: I am highly willing to increase the score if this issue is explained thoroughly.)
>
> **A:** Thanks for your question! The virtual configuration optimization strategy in our UniDrive framework is effective, as demonstrated by the experimental results. We answers detailed questions below.
>
> >**Q1.1:** In certain camera configurations, the results from a single dataset in the paper are not as good as those of the original BEVFusion-C (e.g., 6x60 69.3 vs. 64.6, 6x80b 69.1 vs. 63.1). Why can't the virtual configuration optimization improve to achieve a virtual camera space as good as or better than the original BEVFusion-C?
>
> **A:** Thanks for your question! To clarify, the problem our paper aims to solve is the generalization capacity of models transferred between varied camera configurations, rather than the performance on the identical camera configurations. Our optimization strategy is specifically designed to improve this generalization.
>
> Since the virtual projection would introduce information losses, our method theoretically cannot do better than BEVFusion-C (e.g., 6x80b, 6x70, 6x60) for individual inference. Our model achieves the best overall performance accross different configurations though. We also observe experimental outperforming on certain configurations. The reasons are as follows:
>
> 1. **Inconsistent intrinsics** would introduce performance degradation (Section 4.4). We found that even for the original BEVFusion-C, when tested on identical configurations, the performance of 5x70+110 was significantly lower compared to 6x80 and 6x70. Our virtual projection method eliminates the impact of inconsistent intrinsics, which is why it achieves better performance on 5x70+110.
>
> 2. **Impact of camera numbers.** Our virtual projection uses 6 virtual camera planes. This provides richer information compared to setups with only 4-5 actual cameras, thus enhancing performance on 5x75 and 4x95.
>
> >**Q1.2:** In some camera configurations, the results of individual optimization are not as good as those from transfer learning, such as 60x80b (train on 60x80b 63.1 vs. train on 60x70 67.9). Why can't the virtual configuration optimization achieve a virtual camera space as good as or better than the transfer setting?
>
> **A:** Thanks for your question! Our optimization strategy is designed to find a virtual camera plane that balances and minimizes the projection errors from all configurations. In our experiments, we perform virtual projection for **both identical-configuration inference** (e.g. train and test on 60x80b) **and cross-configuration inference** (e.g. train on 60x80b, test on 60x70). Our experimental results show that the average performance for transfer inference is improved significantly, which addressed our problem setting well.
>
> To address your concern, for most circumstances, the identical-configuration performance is better than the transfer inference. However, the identical-configuration inference will also be impacted by the projection errors, and it is possible that the projection errors in identical-configuration inference might be larger than those of some transfer inference settings (e.g., the virtual camera configurations are more closed to the transferred configuration), leading to a performance drop.
>
> >**Q1.3:** For different camera configurations, what do the optimized Virtual Cameras look like?
>
> **A:** Thanks for your questions! We **add** visualizations of the optimized Virtual Cameras in **Figure 4**. The optimized camera parameters are as follows.
>
> | # of Camera | x | y | z | yaw | fov |
> |:-:|:-:|:-:|:-:|:-:|:-:|
> | 1 | 1.50 | 0.00 | 1.50 | 0 | 72 |
> | 2 | 1.10 | -0.40 | 1.50 | -59 | 72 |
> | 3 | 1.25 | 0.50 | 1.50 | 55 | 72 |
> | 4 | 0.48 | 0.00 | 1.50 | 180 | 72 |
> | 5 | 0.75 | 0.43 | 1.50 | 120 | 72 |
> | 6 | 0.80 | -0.40 | 1.50 | -122 | 72 |
>
> >**Q1.4:** How well does the original view warp to the virtual view? Are there any areas that are not properly warped? Can you provide visual results?
>
> **A:** Thanks for your question. Overall, the warping from the original view to the virtual view is highly accurate. The majority of the scene is seamlessly projected. We **add** visualizations in the Appendix **Figure 7**.

---

> ### Author Response · Authors · 2024-11-24
> **Response to Reviewer 3zcR (Part 2/2)**
>
> >**Q2:** While BEVFusion is popular, the paper focuses on vision-centric autonomous driving. It would be beneficial to include experiments with popular detectors in the camera domain, such as BEVFormer and PETR. These differ from methods like BEVFusion-C and BEVDet, as they are based on query-based transformer solutions.
>
> **A:** Thanks for the constructive suggestion. We agree that evaluating our framework on additional methods would further demonstrate its generalizability. We previously prioritized BEVFusion-C as our baseline method as it introduced an efficient BEV pooling operator and runs faster than the other methods, making it widely adopted in real-world applications. Due to the limited time in the rebuttal, we have conducted experiments with BEVFormer as shown below.  UniDrive demonstrated promising adaptability, we will extend to the other models including PETR.
>
> |  | 6x60 (train) | 6x80a | 6x80b | 6x70 | 5x70 + 110 |
> |---|:-:|:-:|:-:|:-:|:-:|
> | BEVFormer | 71.8 | 4.2 | 6.3 | 29.6 | 12.5 |
> | Ours | 69.1 | 66.5 | 69.4 | 67.3 | 63.1 |
>
> |  | 6x70 (train) | 6x80a | 6x80b | 6x60 | 5x70 + 110 |
> |---|:-:|:-:|:-:|:-:|:-:|
> | BEVFormer | 72.1 | 16.4 | 21.9 | 8.5 | 60.5 |
> | Ours | 68.7 | 64.9 | 63.7 | 65.2 | 63.9 |
>
> >**Q3:** Were the number of virtual cameras optimized together?
>
> **A:** Thanks for your question! The number of virtual cameras are not optimized together. In our experiments, we only consider the situation of 6 virtual cameras. Considering the increased workload for optimization and benchmark experiments with camera numbers, we will do our best to provide the results before the discussion ends.
>
> >**Q4:** The paper discusses that Inconsistent Intrinsics have a significant impact. Since real autonomous vehicles also have inconsistent heights, could you include an experimental analysis?
>
> **A:** Thanks for your question! We conduct experiments on identical configurations as follows. Overall, the inconsistent heights can impact performance, but the impact on performance is not as significant as the changes in intrinsic parameters.
>
> |  | 6x1.8 | 4x1.6 + 2x1.8 | 3x1.6 + 3x1.8 | 4x1.8 + 2x1.4 |
> |---|:-:|:-:|:-:|:-:|
> | BEVFusion-C | 68.4 | 67.5 | 66.2 | 65.9|
> | Ours | 67.7 | 67.0 | 67.8 | 66.3 |
>
> We also conduct transfer experiments.
>
> |  | 6x1.8 (train) | 4x1.6 + 2x1.8 | 3x1.6 + 3x1.8 | 4x1.8 + 2x1.4 |
> |---|:-:|:-:|:-:|:-:|
> | BEVFusion-C | 68.4 | 61.5 | 63.9 | 62.5|
> | Ours | 67.7 | 65.3 | 64.8 | 67.9 |
>
> >**Q4.1:** Were the heights of virtual cameras optimized together?
>
> **A:** Thanks for your question! Heights are optimized together. However, for cameras within the same multi-camera configuration, the heights are consistent across all cameras.
>
> >**Q5:** Given the various camera configurations, would training on datasets that mix various configurations to optimize a unified virtual camera space lead to improved results?
>
> **A:** Thanks for your insightful question! Mixing various camera configurations for training might not lead to improved performance. Even when different camera parameters appear on the same vehicle, we observe a slight performance degradation (as we discussed in Q4). However, it remains unclear whether the performance will improve when the dataset size becomes sufficiently large. If time permits, we will include experimental results to address this question before the discussion session ends. We appreciate your patience and interest in this aspect!
>
> >**Q6:** All experiments are conducted on simulation data. Incorporating transfer experiments with real datasets would be more convincing, like the real dataset transfer experiments conducted in 3D Domain Generalization/Adaptation.
>
> **A:** Thanks for your insightful comment! We acknowledge that Incorporating experiments with real datasets would be more convincing. However, our experiments require identical scenes across varying camera setups. This controlled setup is critical for validating our approach and ensuring that our results are truly representative of cross-configuration performance. Since no publicly available datasets provide identical scenes with varying camera setups and collecting such real-world data goes far beyond the scope of this paper, we currently use CARLA as our platform.
>
> |  | nuScenes | Waymo | nuScenes → Waymo | Waymo → nuScenes |
> |---|:-:|:-:|:-:|:-:|
> | BEVFormer | 43.5 | 44.0 | 4.2 | 2.8 |
> | Ours | 42.6 | 41.9 | 26.8 | 18.3 |
>
> To address the reviewer's concern, we provide some experimental results on real-world datasets as follows. While it is challenging to fully decouple the impact of scene variations on the experiments, our method has shown promising effectiveness to some extent, indicating its potential applicability.
>
> ___
>
> We sincerely thank you again for the valuable comments provided during this review.

---

> > ### Comment · Reviewer_3zcR · 2024-11-25
> >
> > Thank you for your insights and efforts. I have already raised my scores.

---

> ### Author Response · Authors · 2024-11-25
>
> **Dear Reviewer `3zcR`,**
>
> Thank you so much for your positive recognition and for raising your scores! We sincerely appreciate the time and effort you invested in reviewing our work, as well as your thoughtful feedback that helped us improve our submission.

---

### Official Review · Reviewer_rjYz · 2024-11-04

**Soundness:** 2
**Presentation:** 3
**Contribution:** 2
**Rating:** 6
**Confidence:** 3

**Summary:**

The paper addresses the challenge of achieving consistent 3D perception in autonomous driving across different camera configurations. It proposes the UniDrive framework, which creates a unified virtual camera space using ground-aware projections and configuration optimization to reduce projection errors. Experiments in simulation demonstrate that UniDrive improves perception robustness across varied camera setups with minimal performance loss.

**Strengths:**

The paper addresses a valuable and relevant problem in autonomous driving, focusing on camera configuration generalization.

The proposed approach sounds generally reasonable, employing a unified virtual camera space and ground-aware projection to help manage variability in camera setups.

Experimental results show clear performance gains, with UniDrive maintaining high perception accuracy across diverse camera configurations and outperforming baseline models.

**Weaknesses:**

W1: The innovation is somewhat limited, as the framework mainly leverages established virtual projection and optimization techniques without presenting substantially novel concepts.

W2: The approach relies heavily on simulated environments (e.g., CARLA), raising concerns about its applicability and robustness in real-world conditions.

**Questions:**

To what extent does the reliance on simulated data affect the generalizability of the results to real-world settings, particularly in urban driving scenarios?

Does the framework introduce any latency or computational overhead that could impact its feasibility for real-time applications in autonomous driving?

---

> ### Author Response · Authors · 2024-11-23
> **Response to Reviewer rjYz (Part 1/2)**
>
> **Dear Reviewer `rjYz`,**
>
> Thanks for devoting time to this review and providing valuable comments. Below, we provide detailed responses to your comments and concerns. The revised manuscript has been uploaded, with all changes clearly highlighted in red for your convenience.
>
> ---
>
> >**W1:** The innovation is somewhat limited, as the framework mainly leverages established virtual projection and optimization techniques without presenting substantially novel concepts.
>
> **A:** Thanks for this comment! We appreciate the reviewer's feedback regarding the perceived limitations in innovation. However, we would like to clarify several key contributions of our work:
>
> 1. **The research problem we address is novel and real-world demanding.** Specifically, we focus on the generalization capability of models trained on one camera configuration and deployed/transferred on a different one. To the best of our knowledge, **no prior work** has systematically explored this cross-configuration transferability.
> 2. We introduced a ground-aware depth assumption technique that enables effective virtual projections **without requiring depth priors**.
> 3. We developed a comprehensive benchmark designed to systematically study the problem of cross-configuration generalization. This **benchmark** provides a **standardized** way to evaluate and compare methods in this new research area.
> 4. The optimization strategy we developed is **specifically tailored to our problem setting.** It addresses the challenges inherent in cross-configuration deployment.
>
> These contributions, taken together, provide a substantial step forward in addressing the challenges of deploying vision-based models across varying sensor configurations.
>
> ---
>
> >**W2:** The approach relies heavily on simulated environments (e.g., CARLA), raising concerns about its applicability and robustness in real-world conditions.
>
> **A:** Thanks for your insightful comment! We acknowledge the reviewer's concerns regarding simulation. However, the use of simulation is not merely to simplify the experiments, but rather it is essential for ensuring identical scenes across varying camera setups without interference from other factors. This controlled setup is critical for validating our approach and ensuring that our results are truly representative of cross-configuration performance.
>
> Since no public datasets provide identical scenes with varying camera parameters and collecting such real-world data goes far beyond the scope of this paper, we use CARLA as our platform, which is broadly used in the Autonomous Driving community [1-3].
>
> [1] B. Zhang, X. Cai, J. Yuan, D. Yang, J. Guo, X. Yan, R. Xia, B. Shi, M. Dou, T. Chen, S. Liu, J. Yan, and Y. Qiao, ReSimAD: Zero-Shot 3D Domain Transfer for Autonomous Driving with Source Reconstruction and Target Simulation, In ICLR 2024.
>
> [2] H. Shao, Y. Hu, L. Wang, G. Song, S. L. Waslander, Y. Liu, and H. Li, LMDrive: Closed-Loop End-to-End Driving with Large Language Models, In CVPR 2024.
>
> [3] Q. Li, X. Jia, S. Wang, and J. Yan, Think2Drive: Efficient Reinforcement Learning by Thinking with Latent World Model for Autonomous Driving (in CARLA-v2), In ECCV 2024.
>
> To address the reviewer's concern, we provide some experimental results on real-world datasets as follows. While it is challenging to fully decouple the impact of scene variations on the experiments, our method has shown promising effectiveness to some extent, indicating its potential applicability.
>
> | | nuScenes | Waymo | nuScenes → Waymo | Waymo → nuScenes |
> |:-:|:-:|:-:|:-:|:-:|
> | BEVFormer | 43.5 | 44.0 | 4.2 | 2.8 |
> | Ours | 42.6 | 41.9 | 26.8 | 18.3|
>
> ---
>
> >**Q1:** To what extent does the reliance on simulated data affect the generalizability of the results to real-world settings, particularly in urban driving scenarios?
>
> **A:** Thanks for your insightful question!
>
> 1. The primary limitation of CARLA lies in the lower complexity of traffic participant behavior compared to the real world, which can impact autonomous driving decision-making evaluation. However, CARLA's sensor physics closely aligns with those of the real world. This ensures that **the generalizability impact** on our **cross-camera configuration** problem setting is **minimal**.
> 2. We used 6 maps in CARLA, ensuring the simulated scenarios were diverse, covering a wide range of **urban driving conditions**, including intersections, pedestrian crossings, varying traffic densities, and occlusion cases.
> 3. CARLA provides a photo-realistic environment with accurate physics, which is recognized and broadly used within the Autonomous Driving community, as referenced in the above citations.
>
> ---

---

> ### Author Response · Authors · 2024-11-23
> **Response to Reviewer rjYz (Part 2/2)**
>
> >**Q2:** Does the framework introduce any latency or computational overhead that could impact its feasibility for real-time applications in autonomous driving?
>
> **A:** Thanks for your insightful question! Basically, our framework does not introduce latency to the driving perception model.
>
> 1. Our framework introduces a plug-and-play module. This module operates at the data input stage of the model as a preprocessing step, which does not affect the neural network's training and inference.
> 2. We implemented our module using CUDA, which ensures high computational efficiency and fast runtime performance.
> 3. The optimization is designed to find the optimal virtual camera parameters and is entirely independent of the model’s deployment. It does not interfere with model loading, training, or inference.
> 4. To further validate the feasibility of our framework, we provided a comparison of inference speeds. We tested on one RTX 4090 GPU.
>
> | | Latency (ms) | FPS (Hz) |
> |:-:|:-:|:-:|
> | BEVFusion-C	| 81| 12.3 |
> | Ours | 88 | 11.7 |
>
> ---
>
> We sincerely thank you again for the valuable comments provided during this review.

---

> ### Comment · Reviewer_rjYz · 2024-12-01
>
> Thank you for your further explanation. I have raised my scores.

---

> ### Author Response · Authors · 2024-12-01
>
> **Dear Reviewer `rjYz`,**
>
> Thank you so much for your positive recognition and for raising your scores! We sincerely appreciate the time and effort you invested in reviewing our work. Thank you once again for your insightful feedback that helped us improve our submission!

---

### Official Review · Reviewer_sBiy · 2024-11-09

**Soundness:** 3
**Presentation:** 3
**Contribution:** 3
**Rating:** 6
**Confidence:** 4

**Summary:**

The paper addresses the challenge of maintaining robustness in perception algorithms for autonomous driving systems as camera configurations—such as intrinsics, extrinsics, and the number of cameras—evolve. The authors introduce a novel re-mapping approach that transforms pixels from the original camera view to a unified virtual view, allowing training and inference to be conducted in this virtual space. This approach improves robustness across different configurations and integrates seamlessly into existing perception pipelines.
In addition, the paper proposes a virtual camera configuration optimization method, enabling the identification of an optimal virtual configuration based on real-world multi-camera systems to further enhance robustness across configurations.
Experimental results on synthesized data show that the method not only enhances cross-configuration performance but also unexpectedly boosts performance on the same configurations.

**Strengths:**

- This paper addresses a valuable real-world demand. As autonomous driving sensor configurations (e.g., camera intrinsics, extrinsics, and the number of cameras) evolve, it becomes crucial to ensure the perception algorithms are robust to such changes.
- The authors propose a re-mapping approach that transforms pixels from the original view into a unified virtual view, enabling training and inference to be conducted on this virtual view. This approach enhances robustness to cross-configuration shifts in camera systems. Furthermore, the method is modular, making it straightforward to integrate into existing perception pipelines.
- A virtual camera configuration optimization method is also introduced, which can quantitatively find a better virtual camera configuration based on a set of real-world multi-camera system to enhance the cross-configuration robustness and perception performance.
- Experimental results demonstrate that the proposed approach significantly improves robustness across different configurations. Interestingly, performance also improves on configurations identical to the training setup. This is surprising, as one might assume additional transformations could lead to information loss rather than enhancement. This raises questions about whether these improvements are dataset-dependent (I posed in Weaknesses).
- The paper presents a well-formulated problem and demonstrates effective solutions under simulated data conditions.

Despite some limitations, especially those related to simulated data, this paper introduces compelling ideas, offering a well-defined problem formulation that could benefit the autonomous driving industry and potentially embodied AI.

**Weaknesses:**

- As the authors noted, one main limitation is that all analyses were conducted in a simulated environment. While this controlled setting is useful for evaluating the method’s potential, it raises concerns about real-world applicability, as training-based detectors are often sensitive to dataset-specific factors. Given the paper’s real-world motivation, validation on actual datasets would be valuable.
- As I posed in Strengths, it is a little counterintuitive that the transformation from the original view to a virtual view may cause information loss but can bring improvements. The authors should consider proposing hypotheses to explain why this transformation improves performance.

Apart from the issues I posed above, I also have some suggestions that will not prevent this paper to be accepted if unresolved. In real-world applications, there are several datasets of different camera configurations. As the proposed method can leverage the virtual view to unify these different configurations, I am curious whether the proposed method can leverage these datasets to train a better-to-all detector on each dataset.

**Questions:**

Please check weaknesses.

---

> ### Author Response · Authors · 2024-11-23
> **Response to Reviewer sBiy**
>
> **Dear Reviewer `sBiy`,**
>
> Thanks for devoting time to this review and providing valuable comments. Below, we provide detailed responses to your comments and concerns. The revised manuscript has been uploaded, with all changes clearly highlighted in red for your convenience.
>
> ---
>
> >**W1:** As the authors noted, one main limitation is that all analyses were conducted in a simulated environment. While this controlled setting is useful for evaluating the method’s potential, it raises concerns about real-world applicability, as training-based detectors are often sensitive to dataset-specific factors. Given the paper’s real-world motivation, validation on actual datasets would be valuable.
>
> **A:** Thanks for the valuable comment. We only used simulated data to **isolate** the effect of camera configurations since we can control the scenes to be exactly the same across different camera configurations.
>
> However, for real-world evaluation, it is not possible to obtain **identical scenes** across varying camera configurations. This **controlled setting** is critical for validating our approach and ensuring that our results are truly representative of cross-configuration performance.
>
> Still, we have conducted some preliminary experiments on real-world datasets as follows. While it is challenging to fully decouple the impact of scene variations on the experiments, our method has shown promising effectiveness to some extent, indicating its potential applicability.
>
> |  | nuScenes | Waymo | nuScenes → Waymo | Waymo → nuScenes |
> |---|:-:|:-:|:-:|:-:|
> | BEVFormer | 43.5 | 44.0 | 4.2 | 2.8 |
> | Ours | 42.6 | 41.9 | 26.8 | 18.3 |
>
> ---
>
> >**W2:** It is a little counterintuitive that the transformation from the original view to a virtual view may cause information loss but can bring improvements.
>
> **A:** Thanks for the question. Due to the information loss introduced by the virtual transformation, there is indeed a slight performance degradation sometimes when evaluated on identical training configurations (e.g., 6x80b, 6x70, 6x60).
>
> However, our work mainly aims to enhance **model generalization**. Specifically, the main objective is to ensure that a model trained under one camera configuration can be deployed/transferred to other configurations while maintaining similar performance.
>
> Our method shows an improvement in average performance on **transfer tasks** across different camera configurations. We also observe performance improvements on certain configurations (identical to the training setup). The reasons are as follows:
>
> 1. **Inconsistent intrinsics** would introduce performance degradation (Section 4.4). We found that even for the original BEVFusion-C, when tested on identical configurations, the performance of 5x70+110 was significantly lower compared to 6x80 and 6x70. Our virtual projection method eliminates the impact of inconsistent intrinsics, which is why it achieves better performance on 5x70+110.
> 2. **Impact of camera numbers.** Our virtual projection uses 6 virtual camera planes. This provides richer information compared to setups with only 4-5 actual cameras, thus enhancing performance on 5x75 and 4x95.
>
> ---
>
> > Apart from the issues I posed above, I also have some suggestions that will not prevent this paper to be accepted if unresolved. In real-world applications, there are several datasets of different camera configurations. As the proposed method can leverage the virtual view to unify these different configurations, I am curious whether the proposed method can leverage these datasets to train a better-to-all detector on each dataset.
>
> **A:**  Thanks for your constructive suggestion! We appreciate the idea of leveraging real-world datasets with different camera configurations to train a unified detector using our proposed method. However, due to time constraints, we are currently unable to explore this direction within the scope of this submission. Utilizing multiple datasets with different camera configurations has the potential to improve performance. However, increasing the number of datasets can also increase projection errors due to greater variability across camera configurations. This highlights the importance of our **optimization** process, which is designed to find the optimal virtual camera space. We believe this is a promising avenue for future research.
> Thank you again for your insightful feedback!
>
> ---
>
> We sincerely thank you again for the valuable comments provided during this review.

---

### Author Response · Authors · 2024-12-04
**Summary of Author-Reviewer Discussion**

**Dear Reviewers, Area Chairs, and Program Chairs,**

As the Author-Reviewer Discussion session comes to a close, we would like to express our sincere gratitude for your time, effort, and thoughtful feedback throughout this review process.

We are pleased that our rebuttal and revisions have addressed many of the reviewers’ concerns, and we appreciate the positive feedback and insightful suggestions we received.

---

We would like to highlight again the **technical contributions** of this work:

- To the best of our knowledge, UniDrive presents the first comprehensive framework designed to generalize vision-centric 3D perception models across diverse camera configurations.
- We introduce a novel strategy that transforms images into a unified virtual camera space, enhancing robustness to camera parameter variations.
- We propose a virtual configuration optimization strategy that minimizes projection error, improving model generalization with minimal performance degradation.
- We contribute a systematic data generation platform along with a 160,000-frame multi-view dataset, and benchmark evaluating perception models across varying camera configurations.

---

We are encouraged that our reviewers recognize this work:

- **Reviewer `sBiy`:**
    - *"addresses a valuable real-world demand"*, *"presents a well-formulated problem and demonstrates effective solutions”, “enhances robustness to cross-configuration shifts in camera systems"*, and *"benefit the autonomous driving industry and potentially embodied AI."*
- **Reviewer `rjYz`:**
    - *"addresses a valuable and relevant problem in autonomous driving"*, *"proposed approach sounds reasonable"*, and *"show clear performance gains."*
- **Reviewer `3zcR`:**
    - *"greatly appreciate this paper", "enhancing robustness to different camera configurations",* and *"presents a systematic data generation platform and a benchmark."*
- **Reviewer `DD4T`:**
    - *"The article is clear and easy to follow"*, *"… is a long-standing problem in the 3D detection domain, and the authors propose a feasible way"*, and *"model performance … is significantly improved."*

---

As suggested by our reviewers, we have revised the manuscript accordingly. We present a **summary of changes** as follows:

- **Methods & Technical Details:**
    - As suggested by **Reviewer `sBiy`**, we explained reasons for the improved performance with virtual transformation and conducted additional analyses on cross-configuration robustness.
    - As suggested by **Reviewer `rjYz`**, we provided insights into the latency or computational overhead of our method, which shows minimal impact on real-time feasibility.
    - As suggested by **Reviewer `3zcR`**, we visualized the optimized virtual camera parameters, and addressed the core concerns regarding virtual configuration optimization effectiveness.
    - As suggested by **Reviewer `DD4T`**, we provided evaluations on randomly selected configurations to compare with optimized virtual configurations.
- **Experiments:**
    - As suggested by **Reviewer `sBiy`**, we conducted preliminary experiments on real-world datasets, showing the method’s potential applicability beyond simulation.
    - As suggested by **Reviewer `rjYz`**, we included new experimental results to demonstrate the feasibility of our method in real-time settings.
    - As suggested by **Reviewer `3zcR`**, we added experiments for the impact of inconsistent heights.
    - As suggested by **Reviewer `DD4T`**, we performed experiments with popular detectors like BEVFormer.
- **Elaboration:**
    - As suggested by **Reviewer `sBiy`**, we expanded on the practical benefits of unifying camera configurations in real-world applications.
    - As suggested by **Reviewer `rjYz`**, we discussed the limitations of simulation and outlined future directions to incorporate real-world datasets comprehensively.
    - As suggested by **Reviewer `3zcR`**, we clarified the optimization process for virtual camera configuration.
    - As suggested by **Reviewer `DD4T`**, we provided a guide to integrating the method with existing models.

For detailed responses regarding each of the above aspects, please kindly refer to the **comment windows** in the review section.

---

We hope these revisions have strengthened our manuscript and fully addressed your feedback. We are encouraged by the **positive recognition** of our work’s potential impact, as highlighted by **Reviewer `3zcR`**’s comments, who has upgraded the rating from *"6: marginally above the acceptance threshold"* to *"8: accept"*.

We also sincerely appreciate the positive recognition from **Reviewer `sBiy`**, **Reviewer `rjYz`**, and **Reviewer `DD4T`** who rated our work *"6: marginally above the acceptance threshold."*

---

Last but not least, we sincerely thank our reviewers, ACs, and PCs again for the valuable time and efforts devoted and the constructive suggestions provided during this review.


Best regards,

The Authors of Submission 1655

---

### Meta-Review · Area_Chair_NNc9 · 2024-12-23

**Metareview:**

The paper presents a method to improve 3D perception across sensor configurations. The problem is a well-recognized one and the proposed solution, while not significantly novel, is technically sound. The paper is well-written with convincing experiments. Overall, the AC agrees with the reviewers that the paper is ready for acceptance at ICLR and recommends that the authors include rebuttal clarifications in the final version of the paper.

**Additional Comments On Reviewer Discussion:**

Several questions are posed by 3zcR on the virtual configuration optimization, which are effectively addressed by the rebuttal, leading to a positive accept rating. DD4T requires additional results on amera-only 3D detection methods and is also satisfied with the rebuttal response. Concerns on novelty and robustness in real-world settings are raised by rjYz, but also addressed in the rebuttal, causing the reviewer to increase scores towards acceptance. Reviewer sBiy also finds the paper to be interesting and enabling new applications.

---

### Decision · Program_Chairs · 2025-01-22

Accept (Poster)